# Re-Sounding Alarms: Designing Ergonomic Auditory Interfaces by Embracing Musical Insights

**DOI:** 10.3390/healthcare8040389

**Published:** 2020-10-08

**Authors:** Liam Foley, Cameron J. Anderson, Michael Schutz

**Affiliations:** 1Department of Psychology, Neuroscience, and Behaviour, McMaster University, Hamilton, ON L8S 4L8, Canada; andersoc@mcmaster.ca (C.J.A.); schutz@mcmaster.ca (M.S.); 2School of the Arts, McMaster University, Hamilton, ON L8S 4L8, Canada

**Keywords:** medical alarms, auditory alarms, music cognition, music perception, ergonomics, human-factors, sound design

## Abstract

Auditory alarms are an important component of human–computer interfaces, used in mission-critical industries such as aviation, nuclear power plants, and hospital settings. Unfortunately, problems with recognition, detection, and annoyance continue to hamper their effectiveness. Historically, they appear designed more in response to engineering constraints than principles of hearing science. Here we argue that auditory perception in general and music perception in particular hold valuable lessons for alarm designers. We also discuss ongoing research suggesting that the temporal complexity of musical tones offers promising insight into new ways of addressing widely recognized shortcomings of current alarms.

## 1. Introduction

Alarms are effective tools for conveying critical information in cognitively demanding environments. In busy hospitals, construction yards, and airplane cockpits, alarms effectively alert users about crucial information, typically using flashing lights, bright colours, and loud synthesized melodies. Although both visual and auditory alarms can play important roles, auditory signals offer some important advantages—the most obvious being detection even when users are not in the alarm’s line of sight. As they offer faster response times compared to visual alarms [1], auditory alarms provide effective interfaces in saturated environments such as aircraft cockpits, where reliable communication through alarms plays a critical role in ensuring safe operations [2,3]. They are widely used throughout hospitals in intensive care units [4,5,6], neonatal care [7], and operating rooms [8,9,10], among other areas, anywhere medical personnel need to stay apprised of rapid changes in vital signs from numerous patients simultaneously [11].

The acoustic structure of alarms plays an important role in the degree to which they effectively communicate. Although there are numerous ways of classifying alarm sounds, perhaps the clearest delineation is between speech and non-speech. Speech-based alarms are easy to understand and directly signal the alerting issue without using abstract signals [12]. However, non-speech signals offer other important advantages: namely, they are language independent and therefore universal, provide privacy when conveying confidential information, and remain recognizable when speech dominates the ambient noise level [13,14]. The types of non-speech sounds available to designers are vast, ranging from a near-infinite array of synthesized tones to naturalistic referential sounds with clear built-in associations. Consequently the benefits of non-speech alarms generally outweigh those of speech-based alarms—particularly in medical settings, where privacy, universality, and clarity are crucial.

## 2. Medical Device Alarms

Auditory alarms convey a wealth of information to medical personnel, ranging from routine messages related to patient vitals and/or machine maintenance to critical alerts about potentially life-threatening situations. They offer medical professionals a language-independent way of monitoring crucial information in real time, while shielding patients from direct recognition of these messages. They also facilitate communication between staff separated by large distances, such as nurses monitoring patient beds spread across a ward. Given the number of manufacturers of devices used in hospitals throughout the world, message standardization is an important step to improving interoperability (but [15] argues there are still too many alarms despite standardization efforts). Some benefits of interoperability include optimized human–environment interactions (i.e., by applying best design practices determined through rigorous empirical research); [16] and consistent communication between machines and people (by signaling medical events with predictable tone sequences). 

One of the more widely discussed approaches to alarm standardization is the one specified in the International Electrotechnical Commission (IEC)’s global standard, IEC 60601-1-8 [17], which specifies short tone sequences to signal key states at two levels of urgency. Widely adopted, alarms following these guidelines are routinely heard in medical environments across the world. Figure 1 shows eight medium-priority melodic alarm sequences implemented in accordance with the commonly implemented IEC 60601-1-8:2006 standard, depicting the tone relations frequency in Hertz (1a) and in music notation (1b) (Figures can also be seen at Appendix A link). The assumption that adequate pitch or melody discrimination is present throughout the general population may also be shortsighted. For one, pitch and melody discrimination is highly dependent on musical aptitude, with musicians faring better than non-musicians in pitch discrimination tasks [18,19]. As well, certain subpopulations suffer from varying degrees of deficits in the ability to perceive melody or discriminate pitch, namely those suffering from congenital amusia [20,21]. However issues with specialized populations aside, even a cursory musical analysis illustrates numerous potential challenges for anyone trying to differentiate between these sequences in high-pressure situations. For example, the melodies use identical rhythms, similar frequency ranges (all are between 262 and 523 Hz—corresponding to one musical octave), and share a starting pitch. Although ensuring uniformity, these similarities pose serious barriers to effective auditory communication due to frequent confusion [22] and misidentification [23]. Additionally, they suffer from well-known issues related to poor learnability [24] and reduced perceptual clarity [25] as a result of these structural similarities.

As problems with standard alarm sounds are both numerous and well documented (see [13,26] for general discussions of these problems), here we focus on two specific issues: (1) simultaneous masking (when concurrent alarms prevent one another from being heard); and (2) confusions between alarms (when one alarm is mistaken for another). Both reflect auditory challenges faced by musicians for centuries—who have derived useful insights and workarounds enabling effective auditory communication. They are indicative of how lessons learned from music might have prevented what are now widely recognized as shortcomings of current alarm design.

### 2.1. Alarm Masking

Simultaneous masking of auditory alarms occurs when one signal renders another inaudible—an issue frequently cited as problematic in medical environments [25,27,28]. Engineers and alarm designers have long been aware that similarities in acoustic structure increase the risk of simultaneous masking, and suggest including multiple harmonics, as well as intensities and dominant frequencies differing from ambient noise to reduce masking risks [29]. However, additional similarities in an auditory signal’s temporal structure [30], pitch range [31], and timbre [32] also contribute to masking. For example, analyses of signals in the operating rooms and intensive care units report masking of urgent alarms both from high urgency sounds (i.e., other alarms [8]) as well as low urgency sounds (i.e., the telephone [9]). In some contexts, masking reduces alarm audibility as well as recognition [25]. This is obviously problematic in medical environments, where reduced detection and/or increased response times can have dangerous consequences for safety.

### 2.2. Alarm Confusions

Confusions present another problem preventing alarms from realizing their full potential in medical settings. Fewer than 30% of undergraduates can successfully identify alarms from a globally recognized alarm set after training [24]. Other studies on alarm recognition report identification rates of 39% for a sample of nurses [8] and 33% for anesthesiologists [33]. These low recognition rates have major implications for learning. A study examining nurses’ recognition of a universally-standardized alarm set found that only 2 of 14 nurses successfully identified every alarm, with significantly worse recognition for alarms overlapping to any degree [4]. The authors conclude similarities in their sonic properties impair stream segregation—the perceptual ability to distinguish the sources of simultaneous sounds (see [34] for a full discussion of stream segregation)—leading to problematic performance in this important task. It is important to note that this problem has little to do with the medical problems it is signaling, nothing to do with the background of participants, and is unrelated to other problems with alarm fatigue, environmental factors, and issues of noisy environments. Rather it is a reflection of shortcomings in the types of the sounds employed.

In addition to the sounds themselves, musical analysis of the arrangement of sounds within an alarm can explain certain patterns of confusions [22]. For example, the “temperature” alarm featuring the first three notes of a C Major scale (C-D-E) is frequently confused with the “cardiovascular” alarm—featuring the first three notes of a C Major Triad (C-E-G) (Figure 1). As these melodic alarms (i) share the same rhythm, (ii) identical timbres, (iii) use the same rhythm/tempo, (iv) are in the key of C major, and (v) all start on the pitch C, our musical colleagues find challenges with their confusion utterly unsurprising. Unfortunately, what is clearly problematic from a musical perspective is in fact mandated in many guidance documents—such as the IEC standard for melodic alarms. However these challenges are not unique to medical alarms, as insight from music perception is rarely incorporated into alarm design practices.

Together, these well-known issues with both recognition and confusions suggest that greater attention to musical considerations in designing auditory alarms holds significant potential to improve patient care. Although we agree that standardizing acoustic features holds benefits, unfortunately the standards appear to have coalesced around problematic practices generally avoided by musicians. This has in turn led to challenges with both masking [8,9,25] and confusion [18,24]—that are just as problematic as they were predictable. Therefore it behooves designers to contemplate the following. What additional musical insights might inform future alarm design?

## 3. Lessons from Music

Although alarm design is a relatively new field of inquiry, musicians have spent centuries exploring optimal solutions to challenges related to auditory communication—resulting in numerous treatises for composers and arrangers [35,36]. Analyses exploring these principles through a perceptual lens demonstrate they help listeners to perceptually distinguish concurrent voices in busy musical environments [37]. This provides a useful parallel to issues important for alarm recognition in crowded acoustic environments—where both masking and recognition remain serious problems. In some cases, formal research on music perception clarifies and extends our understanding of issues intuitively recognized by generations of musicians. For example, lab-based studies illustrate melodies with a high degree of similarity in pitch range [38] and/or temporal structure [39,40] are poorly discriminated. These approaches provide a controlled way of fully exploring techniques used by composers to avoid confusion in concurrent melodic streams. For example, the heterogeneous use of rhythm and pitch in complex polyphonic music such as Johann Sebastien Bach’s Fugue in B-flat Minor allows listeners to perceptually distinguish five voices performed concurrently on the same keyboard instrument [41] (see [37] for a discussion of grouping principles used in Bach’s polyphonic works). 

### Timbre 

In orchestral performances, the distinctive sound of different groups of musical instruments—such as the brass section or string section—allows listeners to discern their musical roles. This acoustic property encompassing the voice-like quality of instruments and other sounds is called “timbre”. As timbre provides rich acoustic information facilitating differentiation between sound sources, composers frequently use it to offer variations and differentiation between melodies with similarities in pitch and timing. For example, Ravel’s famous Bolero highlights timbre’s salience as a musical cue, [42] enticing listeners to eagerly hear the same two themes repeated nine times each without boredom through masterful manipulation of timbre [43]. Timbre facilitates the recognition of sounds, even for those shorter than a second in duration [44,45] and aids in the perceptual separation of sounds from different sources [30,32,46,47]. In busy settings like the emergency room, this holds the potential to help clinicians discriminate between alarms by differentiating groups of sounds. For example, a study varying alarm timbre found improved recognition among nurses relative to a baseline set of standard alarms [48]. However, unfortunately either through a desire for uniformity or a lack of appreciation for the structure of sounds in music, many alarm standards (such as the IEC 60601-1-8:2006 [17]) mandate the same timbre throughout—precluding the potential benefits of timbral variability [48].

## 4. Sound Issues in Hospitals: What Can (and Cannot) Be Done?

There is a growing understanding of the effects of noise on various areas of public health [14,49,50]. Alarms contribute to problematic noise levels in hospitals—an area of increasing concern as it affects patient recovery time—in part due to sleep loss. Pediatric patients have reported noise levels as a major contributor to sleep disturbances while hospitalized [51]. Lab-based investigations have shown that ICU noise played during sleep significantly increases disruptions [52]. This disturbance in sleep holds important implications even beyond patient comfort, as it is known to affect recovery time [53].

Although simply reducing hospital sound levels would be desirable, much of it is difficult or impossible to mitigate. Some noisy devices are critical (e.g., MRI machines), with their clinical benefits clearly outweighing their acoustic cost [54]. Other environmental sounds such as doors closing, staff conversing, and footsteps resonating on hard floors are natural consequences of environmental constraints (e.g., doors separate rooms; hard surfaces afford better cleaning). However, in contrast to unavoidable noise, alarms represent noise of a different type. As *designed* sounds there is no reason they couldn’t be more optimally created and deployed. Consequently they represent a prime candidate for improving the sonic environment of hospitals—an improvement benefiting both patients and staff. 

One obvious improvement to the amount of noise coming from alarms is to simply reduce their number. However, a “better safe than sorry” [3] design philosophy, in which each individual machine sounds frequently in order to avoid notification of potentially crucial changes inevitably leads to large numbers of alarms [3,27]. Observational studies have found alarm rates exceeding four hundred in forty nine hours, averaging an alarm every seven minutes [55]. One study at John Hopkins Hospital found over three hundred and fifty alarms per patient per day [56], [57] which can lead to medical staff feeling overwhelmed by the cacophony [55,58]. Although decreasing alarm numbers would help address the “acoustic traffic jam” they pose, there is a high medical cost for alarms failing to alert at critical moments. With an ever-increasing number of devices in hospital environments (each with their own alarms) monitoring an increasing number of vital signs and states, the alarm rate is likely to increase in the future [59]. This is similar to trends noted in other industries such as railways [60]. Although music offers little insight into how and when alarms should sound, as a domain built upon auditory communication, it holds powerful insight into best practices for improving the design of the alarm sounds themselves.

### 4.1. On the Complexity of Musical Sounds

Today’s sound designers possess an astonishing array of tools for analysis, synthesis, and manipulation of sound. Although auditory human–computer interfaces are a contemporary problem, some of humanity’s most gifted ears have historically focused intensively on creative approaches to auditory communication—through music. Previously we discussed how music perception research [18,37,38,39,47] offers insight into issues of masking and alarm confusion. Now we turn our focus to musical insights regarding the acoustic structure of the alarms themselves.

Although it might seem too obvious to bear stating, musicians and audiences have long placed a premium on sound. Violins made by esteemed luthier Antonio Stradivari fetch upwards of $3.5 million dollars due in part to prestige, rarity and, above all, sound quality [61]. Adulation of these instruments is so intense that the area in which they were originally made has received a UNESCO Heritage award designation [62]. This devotion to quality sound stems in part from an aspect of alarm design that has often gone underappreciated—sounds’ temporal complexity.

Drawing a bow across a string produces vibrations not only at the fundamental frequency (corresponding to the string’s length), but also at twice that frequency, three times, etc. In musical contexts, these tones over the fundamental are typically referred to as “harmonics”. The harmonics of a typical violin note can be seen in Figure 2a, identified along the x axis (left to right across the page). However, an instrument’s rich sound comes not only from the strength (y axis, bottom to top) of individual harmonics, but how their strength changes over time (z axis, moving “into the page”). These complex temporal dynamics are characteristics of musical sounds, such as notes produced by the trumpet (2b) or clarinet (2c). These changes are part of what makes musical sounds so rich and listenable, so attention grabbing yet pleasing. They are part of what draws us into intense listening without being overwhelmed by the multitude of notes in carefully constructed compositions. Crafted in the collective over centuries of instrument refinement and honed individually through thousands of hours of personal practice, such sounds offer useful insight into ways of structuring auditory messages. Unrestricted by physics and without any need for scarce materials, alarm designers can produce a near-infinite array of tones. Given this universe of possibility, how do typical alarm sounds compare with musical instruments?

### 4.2. On the Simplicity of Alarm Sounds

In contrast to the rich temporal dynamics of musical tones, alarm tones such as those recommended by the 2006 IEC standard are generally temporally simplistic. Similar to musical sounds they contain energy at several harmonics, yet these harmonics exhibit little temporal variation beyond quickly ramped onsets and offsets (Figure 3c). Although quite different than natural musical sounds, it is also understandable given that both music-specific [63] and general auditory [64] perception research overwhelmingly focuses on the kinds of simplistic tones shown in Figure 3a,b. The defining characteristics of these flat tones is their lack of temporal complexity, leading to sounds with decidedly unmusical sounds. Similar to the sounds of a touch-tone phone, these structures follow a precise, easily replicable formula that can be implemented consistently and across many devices. The lack of variation in their temporal structures makes for a certain uniformity—beneficial for considerations such as masking in that the tones have similar amounts of energy throughout. However, the unfortunate lack of validation in current alarm systems [65,66] means they are largely driven by technical consideration rather than perceptual best practices.

In comparison to musical sounds, the temporal complexity of alarm sounds is negligible. Although musical sounds have been carefully sculpted by generations of musicians and audiences to “work” as auditory communication tools, the latter appear designed primarily for the sake of simplicity. It is worth reflecting on the fact that despite generations of intense interest from musicians in exploring sound, and the construction of numerous instruments across the world [67], there has been little interest in temporally simplistic tones. In fact, centuries of refining instrument construction and years of refining individual practice has resulted in musicians continually choosing rich, temporally complex sounds rather than temporally constrained tones [68]. What would happen if alarm design embraced, rather than avoided, this approach?

### 4.3. Temporal Complexity and Annoyance

Our team is currently studying two ways in which music’s use of temporal complexity can improve alarm design: lowering annoyance and improving recognition compared to current alarms. Stimuli in our annoyance experiments consist of tones with the temporal simplicity of normative alarm tones (Figure 3a,b) in contrast with temporally dynamic variations shown in Figure 4a,b inspired by musical tones. These “decaying” tones are based on sounds produced from instruments such as the piano (Figure 4c) or percussion instruments such as the marimba. Our experiments compare the performance of participants hearing either temporally simplistic or dynamic sounds. Across several experiments, participants consistently find the dynamic sounds significantly less annoying then the simplistic tones typically used as the basis for auditory alarms. Yet participants have no more difficulty learning the associations between the alarms and their messages, nor do they perform worse when later asked to recall the alarm sequence’s meaning [69]. These findings are consistent with our team’s previous work demonstrating that tones with complex temporal structures are as easily learned and have the same retention as tones with industry-standard simplistic temporal structures [22].

Our ongoing work on recognition explores the perceptual benefits of “re-orchestrating” standardized alarms with more complex sounds. For example, Figure 5 visualizes three different approaches to synthesizing an alarm signaling power failure of a medical device, with the pitch and timing structure prescribed in the IEC melodic standard. This includes sounds based on (a) the complexity of the violin note shown in Figure 2a, (b) the complexity of the trumpet note shown in Figure 2b, and (c) the simplicity of the standardized IEC tone structure appearing in Figure 5c. Although the instrument-inspired alarm sequences appear more challenging to parse visually, we find their sound sequences easy to grasp auditorily—similar to short three-note melodies played on the violin or trumpet. We are currently planning the formal assessments required to determine the efficacy of such approaches. However, consistent with our previous findings regarding annoyance, our pilot work suggests this approach holds great potential to reduce alarm annoyance without harming learning, recognition, or detectability [69].

### 4.4. Developments in Alarm Standards.

With a growing recognition of the problems associated with discrimination and recognition of the current medical alarm standard, the IEC amended the standard in 2020 [70]. Unlike the 2006 IEC alarms, this standard has been designed in accordance with auditory perception principles and extensively validated [70]. Each alarm consists of an auditory icon—a sound with a real-world referent [71]—representing the alarm source (cardiovascular, profusion, power failure, etc.) paired with an auditory pointer indicating its urgency level. The auditory pointers are similar to alarms specified in the previous standard. They are temporally simple with relatively few harmonics. The alarm uses a single note to denote low urgency, three notes to denote medium urgency, and a pattern of ten notes to denote high urgency. This approach results in sounds with much more temporal variation, with the hope of increasing learnability and detectability.

The extensive validation and evidence-based design and implementation represents a significant advancement over the previous standard [65,66,70]. This validation explores important perceptual considerations related to detectability, resistance to masking, urgency (in the case of the auditory pointers), and learnability [70]. Although this approach of validating is a welcome step, further investigation can lead to subsequent improvements in issues that have not been widely explored. For example, alarm annoyance has not yet been fully considered, yet it plays a major role in alarm ergonomics and shaping the soundscape of medical environments. Consequently we will turn our focus here in the paper’s final section, highlighting where insights from music perception hold great potential for future directions in alarm design.

## 5. The Future of Alarms

The importance of auditory ergonomics can perhaps be seen most clearly by examining the consequences of having ignored such issues previously. During early efforts to create effective alarms in the 1980s, a national standards group turned to well-known acoustician Roy Patterson for assistance. An expert in alarm design, he created a series of sounds designed to provide efficient means of human–computer interactions. Unfortunately these alarms—grounded in his deep knowledge of auditory perception—struck some medical professionals as off-putting [66]. Consequently, they were abandoned in favour of alarms designed by anesthesiologists attempting to create less aversive signals—without considering basic principles of music perception. Their efforts led to the 2006 IEC standards, along with its well-known issues related to masking [25,27,28], challenges with recognition [4,8,33] and problems with confusion [18,24] summarized previously in Section 2. When the shortcomings of that standard became clear, Frank Block—a member of their development team—personally apologized for making medical staff “suffer… with poorly-designed alarm sounds” [66]. As this history is well documented elsewhere, for our purposes, we note merely that rejection of Patterson’s alarms came not from concerns with their detectability, efficacy, or technical feasibility, but rather from concerns with their aesthetics—that they sounded like “a set of random electronic noises” [66]. 

We believe consideration of aesthetics represents an underexplored aspect of alarm design. Although their primary purpose can (and must) be communicative efficacy, their effectiveness cannot be fully separated from the consequences of long-term exposure to aversive sounds. Alarms that might meet useful benchmarks for detectability, masking, confusion, etc., will not be as effective in practice if they are problematic and annoying when heard for hours on end. Much as there are negative consequences from the lack of variability and heterogeneity in current alarms, there can be negative consequences for ignoring many of the issues related to effective acoustic communication—an area explored extensively (if not explicitly), by musicians. Consequently, music perception research and lived musical experience together offer valuable insight into best practices for designing effective, efficient, and ergonomically-sound auditory interfaces—particularly in noisy, high-consequence environments such as ICUs and operating rooms.

Admittedly, challenges with improving alarm efficacy in medical contexts go well beyond their acoustic structure alone. In fact, one of the key issues is simply their prevalence [55,56]. As the majority of alarms in medical contexts are not indicative of a life-threatening situations, the combination of their low predictive value and their problematic acoustic structure leads to situations where simply turning them off can in many cases seem tempting [10,72]—particularly when the sounds used are annoying and off-putting. Unfortunately, these issues will likely only escalate in the future as more devices are developed [59], further exacerbating the problems outlined in this paper.

Alarms will play a major role in shaping the acoustic landscape of medical care for the foreseeable future. Although music perception cannot directly inform how or when to sound these alarms (e.g., device numbers, safe critical thresholds), it offers useful insight to the design of the alarms themselves through centuries of informal trial-and-error experience dealing with perceptual tradeoffs such as detectability vs. annoyance, as well as recognition vs. confusion. Consequently, musical insights can provide useful guidance on auditory ergonomics. As our focus here is on one facet of music—temporal variability—we will now conclude with three main recommendations. 

First, embracing temporal variability can help lower annoyance without sacrificing communication. Our previous and ongoing work shows that increasing the temporal variability in alarm sounds decreases annoyance ratings compared to standard alarm sounds [69]. Importantly, participants detect these alarm sets at the same rate as the standard temporally invariant tones. Intriguingly, although temporal variability serves as both an important tool for future design, it is also possible to introduce this complexity within current constraints on tone pitches and timings, allowing for variations on existing sounds. As they would share the same mapping of melody to alarm source, they would also backwards-compatible with existing standards.

Second, slight changes in timbre such as normal note-to-note fluctuations in musical passages have been shown to enhance detectability in psychoacoustic research [30,73,74]. Musical examples such as *Bolero* highlight how differing timbres can make otherwise repetitive musical phrases engaging for the listener [42]. At a perceptual level timbre aids recognition and separation of sound, which are both of paramount importance for healthcare workers identifying alarms which often occur simultaneously [30,32,44,45,46,47]. Utilizing transient changes in timbre could also improve the communication of existing alarms by reducing their repetitiveness while preserving their recognizability.

Third, the timbres of alarms could be varied by bed, ward, or other section to help improve discrimination and identification. In conjunction with the previous recommendations, using timbre to aid localization could hold important benefits. Recent research suggests manipulations to timbre within certain existing constraints can be a useful mechanism for improving alarms [48]. Future alarm designs could take advantage of the differentiating effect of timbre to better communicate where alarms are occurring—minimizing challenges with locating an alarm’s source in an open room when medical staff are caring for multiple patients concurrently. More varied use of timbre in conjunction with other parameters would also increase accessibility of alarms for those with pitch discrimination deficits, who instead rely on surface features such as timbre to discriminate auditory sources [75].

## 6. Conclusions

Bridging the gap between music, engineering, sound design, and human factors will help propel future alarm standards. Although musical performances and alarm signals serve different purposes, the great care musicians exercise in crafting rich sonic environments can reduce annoyance, masking and confusion. This is supported by a wealth of perceptual studies examining how timbre [44], along with temporal changes in pitch [38], and dynamicity [69], adds pertinent perceptual information to the way we interact with sounds while reducing negative aspects of sound such as annoyance and/or confusions. We believe sound complexly offers a useful step toward improving the audibility problems associated with alarms. 

Music is a domain with an incredible amount of concurrent sound. In a symphony orchestra’s performance, 80–100 musicians simultaneously contribute to an expansive sonic experience, evoking a sense of wonder in listeners [76]. Musical instrument sounds contain tremendous complexity even with single notes, contributing to the rich listening experience characteristic of music. The intuitions of musicians and the careful experimental approaches refined by music perception researchers together offer powerful potential insight into improving the ergonomics of auditory alarms. Listeners routinely “… go through all sorts of trouble to hear human performances rather than the ‘dead-pan’ rendition of computers” [77]. Yet curiously medical alarms typically fixate on simplistic sounds which lead to numerous problems including reports of clinicians turning annoying alarm systems off altogether (see [56,59]).

Luckily, the near-endless range of possible sounds afforded by modern technology—along with new insights from the growing field of music perception research—allow sound designers to devise clever solutions balancing aesthetic and perceptual considerations of auditory signals. We are optimistic this future of alarms will yield more effective and more pleasant auditory signals than those of current standards, improving the alarm scape as well as patient care.

## Figures and Tables

**Figure 1 healthcare-08-00389-f001:**
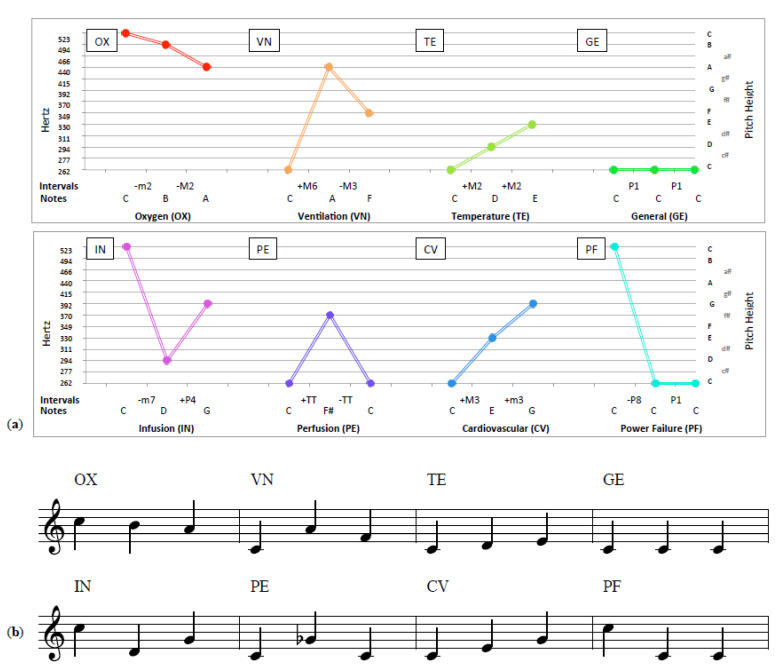
Melodic sequences of the International Electrotechnical Commission 60601-1-8:2006 standard depicted in hertz (**a**) and music notation (**b**). Reprinted from [22].

**Figure 2 healthcare-08-00389-f002:**
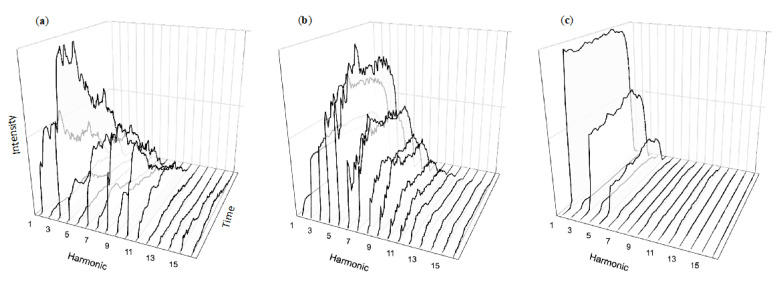
Musical instrument tones feature complex temporal changes simultaneously at multiple harmonics. Although notes from the violin (**a**), trumpet (**b**), and clarinet (**c**) differ in their specific temporal changes, they all exhibit temporal complexity.

**Figure 3 healthcare-08-00389-f003:**
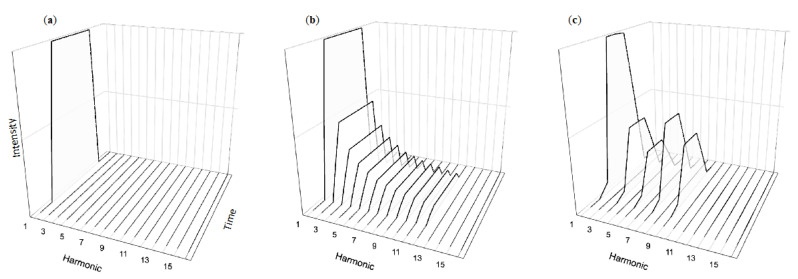
The temporal simplicity of most pure tones (**a**) as well as harmonically rich sawtooth tones (**b**) used in auditory research is similar to those found in industry standard alarm tones (**c**).

**Figure 4 healthcare-08-00389-f004:**
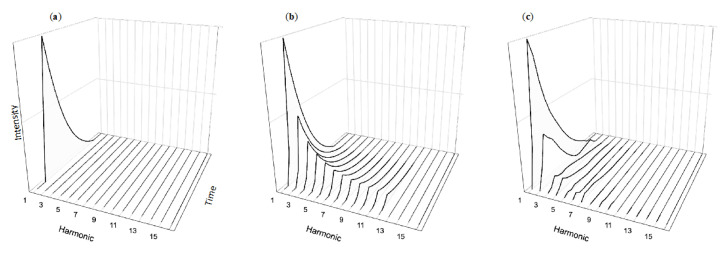
A harmonically simplistic decaying tone, also called a percussive tone (**a**) as well as a harmonically rich percussive sawtooth tone (**b**) and a single piano note (**c**).

**Figure 5 healthcare-08-00389-f005:**
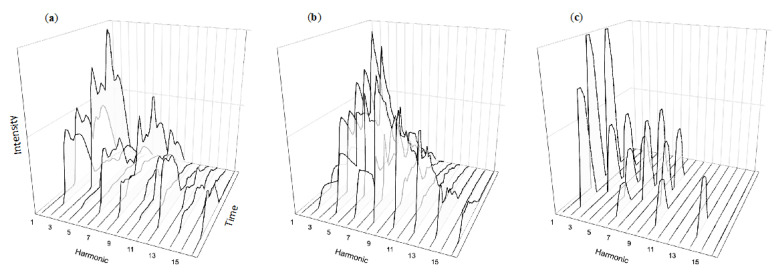
Three distinct renderings of a melodic alarm. The descending octave pitch sequence indicating power failure synthesized with either violin (**a**) or trumpet (**b**) sounds. These exhibit greater temporal complexity than industry standard beeps (**c**).

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
