# Peer review of "Re-Sounding Alarms: Designing Ergonomic Auditory Interfaces by Embracing Musical Insights"

_healthcare, 2020, doi:10.3390/healthcare8040389_

Round 1

Reviewer 1 Report

This is a beautifully written paper that provides a great overview on audible alarms, specifically in the medical domain. Most if not all of the acoustic and psychoacoustic considerations surrounding current generation IEC alarms are covered in this review-style paper, which includes the most important references on research of false alarms, stream segregation, alarm discriminability, simultaneous frequency masking, learnability, and alarm fatigue.  

As an active researcher in the field of audible medical alarms, I would give this review paper to my students as a primer on topic. In particular, I appreciate the discussion of timbre and musicality, as well as the references to Bach and Ravel, as I am based in a Music Engineering program within a School of Music.

One short-coming I would like to point out is that there is no mention of the latest IEC 60601-1-8 amendments that, while not yet completely adopted, have been approved by every voting country and is slated to go into effect in 2020 or 2021 with full normalization by 2024. Importantly, the new guidance moves away from the traditional melodic alarms, and towards a combination of melodic pointers (indicating the imminent annunciation of an alarm, as well as its urgency) in addition to auditory icons. Please see the latest draft here: https://webstore.iec.ch/preview/info_iec60601-1-8%7Bed2.2%7Den.pdf specifically Annex G and H and Table G.

An inclusion and robust discussion of the latest developments of medical auditory alert signals is warranted for a review paper that would be published in 2020 or 2021.

IMPORTANT NOTE TO THE EDITOR AND AUTHORS: I am flagging this as a "Major Revision" only because I would like to see the manuscript again after the additional context has been added. I consider this paper to be relevant to the readership and an important and timely topic. I believe that this paper can be readily and quickly modified into a publishable form, and I urge the Editor to allow extra word count for the authors to make these changes.

Author Response

Thank you for your kind and thoughtful review of our paper.

We completely agree that a mention of the latest IEC 60601-1-8 amendments is warranted. As such we have included a new section on the new standard (4.4 Developments in Alarm Standards).

Thank you and take care,

Liam

Reviewer 2 Report

This is an interesting paper. Its main points are quite timely given the overabundance of distressing noise in today’s communities. The paper is well written and has high readability. The reference list is also quite interesting. The paper itself, however, is rather limited in content and the richness of the reference list is not elaborated in-depth. In its actual format, the paper is also highly speculative and programmatic: it refers to ongoing and future research without giving concrete data. This is the strongest limitation of the paper. Even if there is a reference to previous studies by the authors, some concrete data should at least be given to make the paper more self-containing and to ground the major theoretical claims with empirical findings. I do not know whether there is a word count limit for the paper, but if not, I would recommend elaborating more substantially on the theoretical background of the paper and to provide some more data with regard to the previously conducted research. In order to be as constructive as possible, I list below some general remarks and detailed comments.

General remarks

  • The subject matter of the paper is quite interesting and is very timely.
  • The style of writing is fluent and the English language use is OK.
  • There are some minor spelling mistakes and some grammatical errors.
  • It seems that the methodological part of the study is lacking. There is only reference to previously conducted research, but no data are given. There must at least be some data to make the paper more self-containing.
  • The paper provides interesting perspectives for future research.

Detailed comments

  • line 38: what is meant with opacity? Please explain a little.
  • line 134: Does it make sense to use the numerical ordering 3.1. Is there only one subtitle? The listener should expect here also to find something about the other possible musical parameters.
  • line 151: this sentence is somewhat gratuitous. Can you provide some references to make this more substantial?
  • line 181: tools for analysis. Does this refer to “sound analysis”? Please explain more clearly as there is also the term musical analysis, which is not the same as sound analysis.
  • line 196: Why using ## in Figure ##2a?
  • line 162: multiple spelling mistakes
  • line 246: less annoying than?
  • line 255: our ongoing work on recognition. PIease provide references and, if possible some tentative data.
  • line 257: please explain the term “power failure” a little
  • line 260: Although the the …
  • line 264: our pilot work. Please provide some references or some findings to make this reference stronger.
  • line 271: we have found… Please explain the methodology behind this claim, which now sounds somewhat gratuitous.
  • line 289: foreseeable
  • line 290: spelling mistakes
  • line 295: helps lower…
  • line 296: our previous work. Same remark. Provide references.
  • line 301: explain the concept of backwards-compatibility
  • line 376: the reference is not complete.
  • line 397: why these capitals?
  • line 420: reference is no complete

Author Response

Thank you for your detailed and helpful review of our paper.

In general, we were happy to change the paper as per your comments. However we are unable to provide methods and data from out pilot work here as they are currently under review in a full length empirical article at another journal.  Although we hope to ultimately publish that, it wouldn’t be proper to put the same information and data here.  Additionally, this piece is intended more as a high level description of some useful ideas and concepts to keep in mind in designing future alarms, rather than as a long introduction to a brief discussion of new experimental work.  We see our visualizations of musical tones and alarm sounds as the heart of this paper—essentially the “data” for a manuscript focused on high-level ideas rather than a specific experiment.  Therefore while we appreciate the suggestion and have changed our language to make it clear that… we are not able to add additional experimental information.

Below we answer each of the detailed comments. As some things have moved around some line numbers have changes. So we have changed the number from your original comments when applicable.

line 38: what is meant with opacity? Please explain a little.

We have changed the wording to privacy to be more clear.  The line now reads However, non-speech signals offer other important advantages: namely they are language-independent and therefore universal, provide privacy when conveying confidential information…”

line 137: Does it make sense to use the numerical ordering 3.1. Is there only one subtitle? The listener should expect here also to find something about the other possible musical parameters.

Although we agree with this comment, it seems that this is in keeping with the journal’s numbering conventions but are happy to defer to the action editor’s guidance on this.

line 153: this sentence is somewhat gratuitous. Can you provide some references to make this more substantial?

We agree at and have revised. The line now reads “However, unfortunately either through a desire for uniformity or a lack of appreciation for the structure of sounds in music, alarm standards often mandate the same timbre throughtout the set. One such example is the IEC 60601-1-8:2006,[17], which misses the potential benefits of timbral variability [44].” Here we provide a specific standard which does no differentiate timbre of the alarms, with empircal evidence that doing so, would be beneficial.

line 181: tools for analysis. Does this refer to “sound analysis”? Please explain more clearly as there is also the term musical analysis, which is not the same as sound analysis.

We have clarified this, by adding “of sound” at the end to ensure clarity.  This sentence now reads “Today’s sound designers possess an astonishing array of tools for analysis, synthesis, and manipulation of sound”.

line 240: Why using ## in Figure ##2a?

Deleted the “##”, this was a placeholder from drafting it has been removed.

line 207: multiple spelling mistakes

Fixed spelling mistakes. Now reads “constraints regarding hygiene, alarms represent a different”.

line 246: less annoying than?

Added “compared to current alarms” to ensure clarity. This sentence now reads “Our team is currently studying two ways in which music’s use of temporal complexity can improve alarm design: lowering annoyance and improving recognition compared to current alarms”. 

line 255: our ongoing work on recognition. Please provide references and, if possible some tentative data.

Please see above comment on not including data.

line 257: please explain the term “power failure” a little

We have altered text to clarify. The sentence now reads “For example, Figure 5 visualizes three different approaches to synthesizing an alarm signaling power failure of a medical device with the pitch and timing structure prescribed in the IEC melodic standard”.

line 260: Although the the …

Deleted duplicate ‘the’

line 264: our pilot work. Please provide some references or some findings to make this reference stronger.

We have added a citation for this work.

line 271: we have found… Please explain the methodology behind this claim, which now sounds somewhat gratuitous.

As this claim didn’t relate to explaining the figure, and did need more reference, we have removed the sentence. The figure now reads as “Three distinct renderings of a melodic alarm. The descending octave pitch sequence indicating power failure synthesized with either violin (a) or trumpet (b) sounds.  These exhibit greater temporal complexity than industry standard beeps (c).”

line 460: foreseeable

Typo fixed.

line 456: spelling mistakes

Spelling mistakes have been fixed.

line 295: helps lower

Fixed typo. Now reads “..helps lower”

line 296: our previous work. Same remark. Provide references.

Please see above comment on including methodology and data. We have provided a reference for some of this work.

line 301: explain the concept of backwards-compatibility

We have altered the text to clarify. The line now reads “Intriguingly, although temporal variability serves as both an important tool for future design, it is also possible to introduce this complexity within current constraints  on tone pitches and timings, allowing for variations on existing sounds. These temporally variant alarms would still share the same mapping of melody to alarm source, and thus would be backwards compatible.”

line 441: the reference is not complete.

This has been fixed

line 397: why these capitals?

This has been fixed

line 420: reference is no complete

This has been fixed.

Thank you and take care,

Liam

Round 2

Reviewer 1 Report

Thank you for addressing my questions and concerns. Accept in current form.

Author Response

Thank you for your constructive feedback.